# Real-Time Monitoring the Effects of Storage Conditions on Volatile Compounds and Quality Indexes of Halal-Certified Kimchi during Distribution Using Electronic Nose

**DOI:** 10.3390/foods11152323

**Published:** 2022-08-03

**Authors:** Andri Jaya Laksana, Young-Min Choi, Jong-Hoon Kim, Byeong-Sam Kim, Ji-Young Kim

**Affiliations:** 1Department of Food Biotechnology, University of Science and Technology (UST), Daejeon 34113, Korea; andrijlaksana@kfri.re.kr; 2Enterprise Solution Research Center, Korea Food Research Institute (KFRI), Wanju 55365, Korea; ymchoi@kfri.re.kr; 3Food Safety and Distribution Research Group, Korea Food Research Institute (KFRI), Wanju 55365, Korea; jhkim@kfri.re.kr (J.-H.K.); bskim@kfri.re.kr (B.-S.K.)

**Keywords:** halal-certified Kimchi, storage conditions, volatile compounds, quality indexes, shelf-life

## Abstract

The food logistics system is an essential sector for maintaining and monitoring the safety and quality of food products and becoming more crucial, especially during and after the pandemic of COVID-19. Kimchi is a popular traditional fermented food originally from Korea and easily changes because of the storage conditions. This study aims to evaluate the effects and the contributions of temperature to volatile compounds, quality indexes, and the shelf life of Halal-certified Kimchi, and to identify alcohol and find the correlation between the identified variables using an electronic nose and conventional method with the integration of multivariate analysis. Thirty-two volatile compounds (VOCs) were detected and correlated with pH, titratable acidity (TA), and lactic acid bacteria (LAB) counts during storage time. Ethanol was also found in the ripened Kimchi and possibly became the critical point of halal Kimchi products besides total acidity, pH, and LAB. Furthermore, the correlation between pH and benzaldehyde, titratable acidity and 3-methylbutanoic acid, and among lactic acid bacteria with ethanol, acetic acid, ethyl acetate, and 3-methylbutanoic acid properly can be used as a given set of variables in the prediction of food quality during storage and distribution.

## 1. Introduction

Kimchi is a traditional fermented food, originally from Korea, which can be produced from cabbage (Baechu), green onion, leaf mustard, and radish. It also has unique characteristics of flavor constructed by the combination of the main ingredients and the spices such as chili powder, ginger, green onion, garlic, and salted seafood in the process of making Kimchi that has been applied over a long time [1,2]. Fermentation generally is a biochemical change of organic substances induced by microorganisms or enzymes whether brought by anaerobic or partial anaerobic oxidation [3]. Lactic acid bacteria (LAB), *Leuconostoc gelidum*, *Latilactobacillus sakei*, and *Weisella koreensis*, possibly are major contributors to the quality, safety, and nutrition of the Kimchi during fermentation [4]. This slow decomposition process of organic materials continuously occurs and can highly affect the quality of the final products, whether advantageous or disadvantageous, during distribution and storage [5].

Previous researchers have demonstrated that starter cultures contributed to the production of Kimchi metabolites (organic acids and mannitol), the optimal ripening period, and sensory improvement, and they provide uniform quality for commercial production [6,7]. Other researchers have also investigated and attempted to halt over-acidification using a physicochemical approach by adding the antimicrobial agents and combining them with hurdle effects technology to extend the shelf life of Kimchi [8]. The initial density of microorganisms can be also reduced by 3.29 and 4.78 log cfu/g after implying 10 kGy of gamma irradiation [9]. The combination of high hydrostatic pressure (HHP) treatment and super-cooling storage conditions (−4.5 °C) effectively prolongs the ripening state of leaf mustard Kimchi [5]. Furthermore, the effects of various seasoning ingredients, food additives, and sodium chloride content in the Kimchi have been studied to identify not only the shelf life but also the health effects of the Kimchi diet [10,11,12]. The alteration of the initial temperature of storage to 4 °C for long-term storage (minimum of 31.3 days) with the comparison of one or two days of fermentation at room temperature storage was also conducted [13]. It is believed that temperature control in storage during distribution is the key factor and the easiest practice to control fermentation speed and the formation of early microbes such as *Leuconostoc*, *Weissella*, and *Lactobacillus*.

Aroma-active compounds, such as acetic acid, propionic acid, butanoic acid, 2-methyl propionic acid, hexanal, and ethanol, have been identified specifically as volatile metabolites produced by microorganisms during Kimchi fermentation. These volatile compounds have a positive correlation with lactic acid bacteria growth and can be determined by gas chromatography–mass spectrometry (GC/MS) with the combination of solid-phase micro-extraction (SPME) [14,15,16], vacuum simultaneous steam distillation–solvent extraction/gas chromatography/mass spectrometry (V-SDE/GC/MS) [17], or using GC-MS with an automated purge and trap sampler [18]. An electronic nose based on a fast gas chromatography-flame ionized detector (FGC-FID) system has also the potential to assess food quality (freshness) in the supply chain. The benefit of this method is not only fast measurement but also a small requirement of sample volume [19,20]. Rapid and non-destructive measurements such as Vis/NIR hyperspectral imaging, Fourier-transform infrared spectroscopy (FTIR), and hyperspectral imaging, nowadays, are also becoming more popular especially in the food sector, as they are not only fast in detection but also effective in determining the quality of food products [21,22,23,24]. Furthermore, an electronic nose has been proven and practically can be integrated with various non-destructive techniques for monitoring the food quality and halal status in food products [25,26].

This study aims to evaluate the effects and the contributions of storage conditions, especially storage at room temperature, volatile compounds, quality indexes, and the shelf life of halal-certified Kimchi over storage. Next, it aims to identify alcohol production qualitatively during fermentation in the storage room using an electronic nose and find its correlation with conventional methods of food quality analysis.

## 2. Materials and Methods

### 2.1. Preparation of Samples and Materials

Halal-certified Kimchi, 500 g weight each product, was obtained from a Kimchi manufacturer, and the experiment was conducted immediately after receiving samples at the Korea Food Research Institute (KFRI), Wanju-gun, Republic of Korea. The samples were stored in four storage rooms with different temperature conditions 0 °C, 5 °C, 10 °C, and 20 °C. The sampling procedure was managed variously based on temperature treatment, at 0 °C and 5 °C, and the interval of sampling was 3 days. On the other hand, sampling at 10 °C and 20 °C was performed every day from day 0 until day 6, with a 2-days interval of sampling from day 6 until day 14. The samples were determined by the value of quality indexes (pH, titratable acidity, and lactic acid bacteria count), temperature records, and volatile compounds.

The temperature of the internal product was recorded using a Thermo recorder (TR-5i Series, T&D Corp., Matsumoto, Japan) with two repetitions in each storage room [27]. At the end of the experiment, all data were collected using optical communication port TR-50U2 and displayed through T&D Graph software for data management and visualization.

### 2.2. Analysis of Halal-Certified Kimchi Quality Indexes

#### 2.2.1. pH and Titratable Acidity (TA)

Samples of Kimchi juice (20 mL), which was previously blended using a slow juicer (H-100-SBFA01, Hurom Corp., Seoul, Korea), were placed in a 50 mL beaker glass, and the pH was measured using a digital pH meter (TA-70, DKK-TOA Corp., Tokyo, Japan) [28,29]. 

The titratable acidity was titrated to 20 mL of Kimchi juice by adding 0.1 N NaOH until it reached pH 8.2. The consumed volume of NaOH was calculated and converted as lactic acid content in a percentage (%). The titratable acidity was calculated according to the next Equation (1) [30]:TA (%) = [Volume NaOH (mL) × 90.08 (g) × 0.1 N × 100]/[20 mL × 1000](1)

#### 2.2.2. Lactic Acid Bacteria Count (LAB)

Lactic acid bacteria were counted by the 3M Petrifilm lactic acid bacteria count plate method 6461/6492. The evaluation of Petrifilm is a reliable LAB enumeration method compared with the traditional methodology [31,32]. In sterile conditions, portions of Kimchi (10 g) were obtained from each sample and diluted 10-fold with 90 mL of 0.85% saline solution in a sterile filter bag. After homogenization for 1 min with four strokes/s using a stomacher (BagMixer 400 CC, Interscience Intl., Saint-Nom-la-Bretèche, France), 1 mL of solution was transferred and diluted stepwise with 9 mL of 0.85% saline solution. Afterward, the sample (1 mL) was plated in a 3M^TM^ Petrifilm^TM^ lactic acid bacteria count plate (PLAB) and incubated at 35 °C for 48 h. The counted colonies of the Kimchi sample were expressed as log cfu/g [28].

### 2.3. Analysis of Volatile Compounds

The volatile compound in the sample product was analyzed using GC-FID Heracles II (Alpha M.O.S., Toulouse, France). The instrument consisted of an auto sampling system (HS100 autosampler), two polarity columns, MXT-5 and MXT-1701, and two flame ionization detectors. Five milliliters of Kimchi juice were prepared in a vial and injected into the e-nose in six repeats of the sample at 0 °C, 5 °C, 10 °C, and 20 °C. Before analysis, a method was created with the following parameters: injection volume of 1000 μL with a speed of 125 μL/s at 200 °C of injector temperature, incubation at 40 °C for 20 min with agitation at 500 rpm, and 90 s flushing time between injections. This method was followed by Hydrogen carrier gas flow at 30 mL/min, trapping temperature at 40 °C, initial oven at 50 °C, the endpoint of oven temperature at 250 °C, the heating rate at 1–3 °C/s, an acquisition duration of 110 s, and an acquisition period of 0.01 s. For data calculation, Kovats indices were used to determine the retention time of C6–C16, and the processed-data acquisition was conducted through Alpha-Soft software. Alpha-Soft (V14.2, Alpha M.O.S, Toulouse, France) software is a data acquisition and processing system used for instrument control and raw data processing. The chromatographic results are treated and recorded as input data for chemometric analysis. Sensors (peak areas) with the highest performance power were selected using the function of the Alpha-Soft software. The Heracles e-nose was equipped additionally with the AroChembase (Alpha MOS, Toulouse, France) library which was used for confirming the chemical compound identification and allowed to pre-screen the compounds and sensory attributes from the chromatograms [20].

### 2.4. Bivariate and Multivariate Statistical Analysis

The analysis of variance (ANOVA) was conducted by SPSS statistics 24.0 software (SPSS Inc., Chicago, IL, USA) with Tukey-HSD as a post hoc test at the significance level of 0.05. Unsupervised multivariate chemometric analysis such as principal component (PC) was also performed using R programming version 4.2.0 and RStudio Desktop 2022.02.2 + 485 to calculate and analyze the contribution of temperature, storage period, and other variables of each sample. PCA, or principal component analysis, is a multivariate approach to find new variables that are linear functions of those in the original dataset that successively maximize variance, and that are uncorrelated with each other. PCA is applied to transform a high-dimensional dataset into a lower-dimensional dataset by using the first few principal components [33,34]. Besides, k-means clustering was performed to partition the observations with the nearest mean of volatile compounds and quality indexes of halal-certified Kimchi over storage time. K-means clustering is a distance-based clustering algorithm that aims to minimize the cluster performance index, square-error, error criterion, distances between the points, and their respective cluster centroid [35,36]. The packages used in R were as follows: FactoMineR 2.4 [37], factoextra 1.0.7 [38], ggplot2 3.3.5 [39], and ComplexHeatmap 2.10.0 [40].

## 3. Results and Discussion 

### 3.1. Analysis of Halal-Certified Kimchi Quality Indexes

#### 3.1.1. pH and Titratable Acidity (TA)

Changes in pH during the fermentation of Kimchi indicate the degradation of carbohydrates as the result of microbial activities in the formation of organic acids and other chemical compounds [41,42]. The changes in the pH value of halal-certified Kimchi at different temperatures in the storage room are illustrated in Figure 1a. From the results, the pH value of the halal Kimchi at 0 days or in the fresh condition was 5.96 ± 0.01. The Kimchi stored at 20 °C significantly reduced to a pH of 4.11 ± 0.10 after 2 days, and 6 days of the Kimchi stored at 10 °C reached the pH of 4.13 ± 0.02. On the other hand, the Kimchi stored at 5 °C was relatively stable until 6 days of storage from 5.96–5.55 and was followed by a rapid decrease to 4.50 ± 0.04 in the next 3 days. The Kimchi stored at 0 °C exhibited longer stability conditions of pH compared to the other treatments with a range of 5.96–5.88, with an extreme decline between 15 and 18 days of storage. In the following days, the pH values gradually decreased until 30 days and reached the final value of 4.3 ± 0.02 at 30 days. Another finding showed that the optimal flavor of Kimchi is in the range of 4.0–4.5 [43]. Thus, at 0 °C, the shelf life of the Kimchi was significantly longer (*p* < 0.05), and the ripening stage can be halted in the long-term storage period. 

Titratable acidity (TA) represents the total acidity of lactic acid generated from microorganisms in the Kimchi during fermentation. The percentages of TA in halal-certified Kimchi at different temperatures are presented in Figure 1b. The initial TA of fresh Kimchi (at day 0) showed 0.24 ± 0.01%. After 12 days, the TA of the Kimchi stored at 0 °C gradually increased and reached the maximum value of 0.88 ± 0.00% on day 30. The Kimchi acidity stored at 5 °C did not considerably change after 12 days of storage time with a range of 0.96–1.06%. In contrast, the rapid increase of TA occurred at the beginning of the Kimchi storage at 20 °C from day 1 to 2 with 0.99 ± 0.04%, whereas Kimchi stored at 10 °C increased sharply on day 3 from 0.47% to 0.81% of total acidity as lactic acid. The optimal fermentation in Kimchi occurs at a pH of 4.2 and 0.6% of total acidity [2]. Therefore, the ripening process of halal-certified Kimchi stored at 0 °C possibly can be delayed for a long time, and the results depicted significant differences compared to other treatments (*p* < 0.05). Kimchi at low temperatures (−1 °C) can be kept for 2 to 30 weeks and the acidity is maintained at 0.47–0.50% [44].

#### 3.1.2. Lactic Acid Bacteria (LAB) and Temperature Records during Distribution

The lactic acid bacteria growth at four different temperatures and the history of the Kimchi’s internal temperature are presented in Figure 2. The initial LAB count in the Kimchi was 5.01 ± 0.24 log cfu/g; this result is relatively higher than the finding which exhibited 4.58 log cfu/g of LAB at the beginning of fermentation [45]. The gap number of the microbial count in the Kimchi can be caused by the initial temperature of halal-certified Kimchi in the distribution (8.46 ± 0.84 °C) shown in Figure 2a–d. Different temperatures and packaging conditions (open or closed) could alter the diversity of microorganisms in the Kimchi [46]. In addition, the Kimchi types and the composition of vegetables and seasoning possibly affected the microbial community in determining product qualities during the early phase of fermentation [47].

The LAB count of Kimchi stored at 20 °C drastically increased to 8.34 ± 0.12 log cfu/g in less than 24 h (Figure 2d) and was significantly different compared to other storage rooms (*p* < 0.05) on the first day. The Kimchi stored at 10 °C reached the maximum number of LAB of 8.68 ± 0.05 log cfu/g on day 6, and at 5 °C it was 8.55 log cfu/g on day 12 (Figure 2b,c). In contrast, the Kimchi at 0 °C showed a relatively stable number of LAB (5.01–5.15 log cfu/g) until 6 days of storage and continuously grew to 8.57 ± 0.09 log cfu/g on day 18. The external temperature could likely affect the maximum number of LAB counts in the Spring, Autumn, and Winter seasons with the range of 8.30–8.85 log cfu/g [48]. 

### 3.2. Analysis of Volatile Compounds

All the data of volatile compounds (VOCs) produced during fermentation and their odor descriptors individually are summarized in Table 1. Different from conventional methods, the response should be pre-treated using chemometric analysis with the additional module software from the instrument in order to find the qualitative and quantitative data output and reveal the useful information such as chemical compound, peak area, functional group, and odor descriptor [49]. The changes in the e-nose sensor (chromatogram peak) of the Kimchi stored at 20 °C are provided in Figure 3. It can be seen that during fermentation, the peak of the volatile compound changed gradually from day 0 until day 14. Six major compounds, ethyl isobutyrate, 2,3-dimethyl pyrazine, nonan-2-one, ethanol, ethyl 3-(methylthio)propanoate, and acetaldehyde, were detected in the early fermentation at all temperatures. Each compound represented the sweet, nutty, fresh, alcoholic, sulfury, and pungent nuance of Kimchi, respectively. Changes in the VOCs of the Kimchi varied depending on the functional group of chemical compounds as shown in Figure 4. There were nine groups of VOCs including acids, alcohols, aldehydes, alkenes, benzenes, esters, ketones, pyrazines, and sulfides. Alcohol, esters, and acids in the Kimchi stored at 20 °C rapidly increased compared to the Kimchi at 0 °C, 5 °C, and 10 °C with esters as predominant compounds and followed by alcohols. However, extreme decreases occurred in aldehydes, alkenes, ketones, pyrazines, and sulfides at 20 °C on the first day of storage. Five groups of compounds could be identified in the Kimchi such as alcohol, aldehyde, ketone, ester, and nitrile, of which alcohols were the predominant compounds in the optimum conditions of the fermented Kimchi [50]. Furthermore, thirty-two types of volatile compounds were identified in the fresh and fermented Kimchi during distribution (Table 2). Significant changes, that frankly could be found in Kimchi, were the production of ethyl acetate and butanoic acid at 20 °C (*p* < 0.05). The latter compounds are defined as ethereal, fruity, sweet, acetic, and buttery odors after 14 days of fermentation. Other compounds were also identified at 20 °C in high intensity such as ethanol, butanal, acetic acid, 3-methylbutanoic acid, and 2-heptanone which could give a stronger alcoholic, pungent, sour, stinky, and spicy aroma than the Kimchi at 0 °C, 5 °C, and 10 °C.

### 3.3. Multivariate Analysis

#### 3.3.1. Relationship of Volatile Compounds and Quality Indexes by Cluster Analysis

Clustering is an unsupervised algorithm which calculates the similarity or proximity based on the distance of measurement [51]. Cluster analysis was used to normalize the concentration of volatile compounds by squared Euclidean and applied the standard k-means to a dataset of identified VOCs in the Kimchi. Data mining and clustering related to chemometric databases are common in chemical processing data [52,53]. In the same way, hierarchical cluster analysis can be used to find the correlation between volatile compounds and sensory attributes of traditional sweets [54]. Figure 5 depicts the matrix of the normalized enrichment distance score of 32-volatile compounds and three quality indexes as variables in the function of the correlation between temperature (°C) and storage period (day) by k-means clustering. The variables, measured by e-nose and conventional methods, were divided into two clusters which have different patterns of shifting between increasing and decreasing distances during the fermentation of Kimchi at different temperatures. In general, the Parameter of pH was in the group of negative responses (red dendrogram) during storage together with benzaldehyde, acetaldehyde, and dimethyl trisulfide. By contrast, the titratable acidity and lactic acid bacteria counts were in the positive response group (blue dendrogram) together with 3-methylbutanoic acid. Interestingly, the ethanol content also had a close relationship with the increase in acetic acid, ethyl acetate, 3-methylbutanoic acid, and total acidity and has a negative relationship with acetaldehyde. The increase in ethanol was caused by the conversion of acetaldehyde metabolized by microorganisms [55]. Furthermore, based on the dendrogram tree, the relationship between pH and benzaldehyde, titratable acidity and 3-methylbutanoic acid, and among lactic acid bacteria with ethanol, acetic acid, ethyl acetate, and 3-methylbutanoic acid can be used in the future as a given set of variables to predict a target variable in determining the food quality during storage and distribution using supervised multivariate analysis.

#### 3.3.2. Correlation between Volatiles Compounds and Storage Conditions

Principal component analysis (PCA) is an unsupervised multivariate analysis approach to visualize the correlation between variables in order to find the differences and similarities among data points in the graph [56]. Loading plots and Principal Components (PC)-Biplots of VOCs with PC1 50.8% and PC2 12.7%, and volatile compounds as functional groups with PC1 52.4% and PC2 18.5% are presented in Figure 6. It can be seen in Figure 6A that 2-heptanone, butanal, delta-decalactone, delta-nonalactone, methyl eugenol, geraniol, 2-methylpropanoic acid, and 4-octanolide had a low contribution to the ripening stage of halal Kimchi. On the other hand, the rest of the twenty-four compounds had moderate to high contribution to the fermentation. In addition, Figure 6B shows the relationship of storage temperatures to the identified VOCs during storage in which the fresh Kimchi had a strong concentration in Ethyl 3-(methylthio)propanoate, acetylpyrazine, dimethyl trisulfides, and acetaldehyde (or Aldehyleds and Sulfides in Figure 6D). The halal-certified Kimchi stored at 20 °C produced VOCs separately compared to the Kimchi stored at 0 °C, 5 °C, and 10 °C, which gave a strong ethereal, sour, acetic, sweet, alcohol aroma as represented by ethyl acetate, acetic acid, butanoic acid, 3-methylbutanoic acid, isoamyl acetate, and ethanol (Acids and Alcohols in Figure 6C). In contrast, in the last observation, the Kimchi stored at 0 °C, 5 °C, and 10 °C identically had similar aromas such as sweet, butter, aldehydic, pepper, green, nutty, herbal, peppery, pine, ethereal, floral, slightly bitter, and musty odors produced by nonan-2-one, butane-2,3-dione, n-nonanal, terpinen-4-ol, butyl butanoate, 2,3-dimethylpyrazine, (-)-carvone, L-limonene, 1S-(-)-a-pinen, ethyl isobutyrate, 1-nonanol, benzaldehyde, and 2-propanol, respectively. Allyl mercaptan, methyl allyl sulfide, diallyl sulfide, and ethanol increased in line with the increase in temperature during the ripening process [57]. Thus, the determination of the volatile compound (ethanol) is the important key in halal logistics and distribution, especially in the storage stage, not only to control the quality of Kimchi but also to maintain the validity of halal status in the registered product. 

## 4. Conclusions

Thirty-two volatile compounds, acids and alcohols as the predominant groups, contributed to the qualities and odor of the Kimchi. The changes in temperature conditions also had enormous impacts on the Kimchi’s qualities. The Kimchi stored at 20 °C fast fermented in two days and produced alcohols qualitatively higher than the Kimchi under other storage conditions. Furthermore, a fast and reliable instrument, an electronic nose, could be applied and integrated with conventional methods and multivariate analysis to evaluate the influences of storage conditions and control the qualities (ethanol) of halal-certified Kimchi during distribution. Besides, the relationship between the variables (conventional method and e-nose responses) such as pH with benzaldehyde, titratable acidity with 3-methylbutanoic acid, and among lactic acid bacteria with ethanol, acetic acid, ethyl acetate, and 3-methylbutanoic acid appropriately can be used as a known set of variables to predict a target variable in determining the food quality during storage and distribution.

## Figures and Tables

**Figure 1 foods-11-02323-f001:**
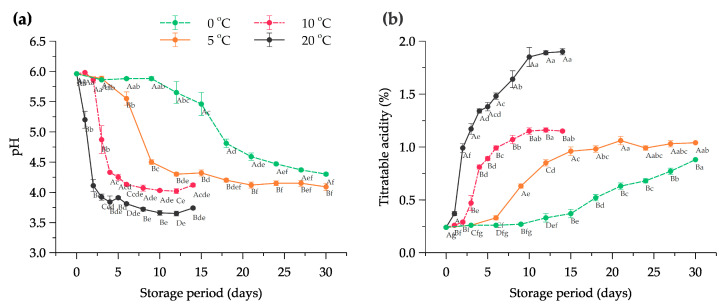
Changes in (**a**) pH value and (**b**) titratable acidity as lactic acid of halal-certified Kimchi stored at different temperature conditions. Note: The data are expressed as mean ± SD with *n* = 3; Superscripts with different letters (A–D) considering on the same day and (a–g) considering at the same temperature condition represent significant differences at the *p* < 0.05.

**Figure 2 foods-11-02323-f002:**
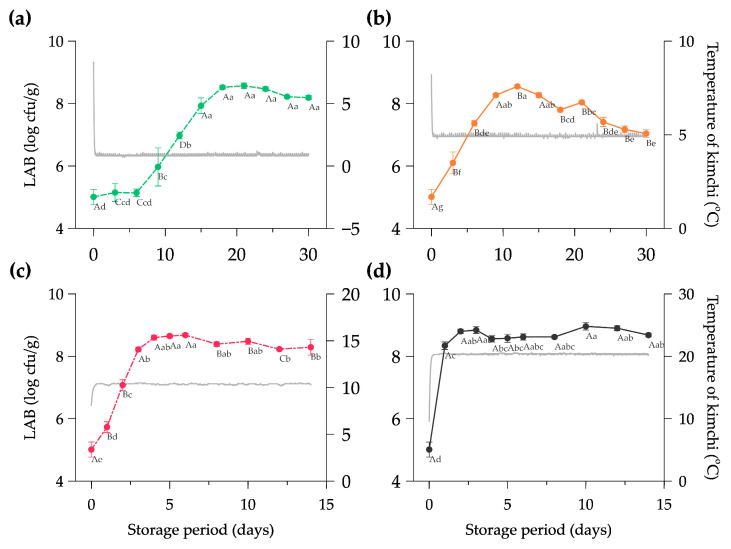
Growth of lactic acid bacteria in halal-certified Kimchi at (**a**) 0 °C, (**b**) 5 °C, (**c**) 10 °C, and (**d**) 20 °C during storage period. Note: Data are expressed as mean ± SD from three repetitions; The mean values highlighted with different capital letters (A–D) considering the same storage time and different lowercase letters (a–g) considering the same temperature treatment are significantly different at the *p* < 0.05; the gray line indicates the evolution of the internal temperature of Kimchi during storage.

**Figure 3 foods-11-02323-f003:**
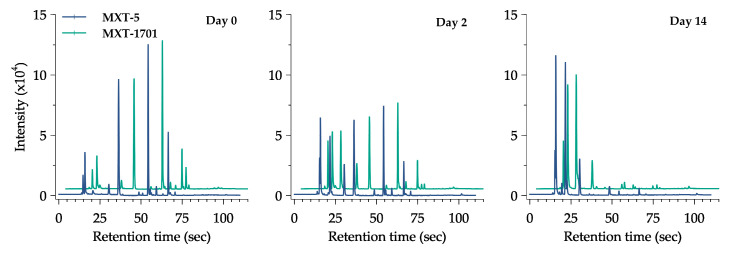
Response change of e-nose chromatogram of halal-Certified Kimchi stored at 20 °C and performed on three different days.

**Figure 4 foods-11-02323-f004:**
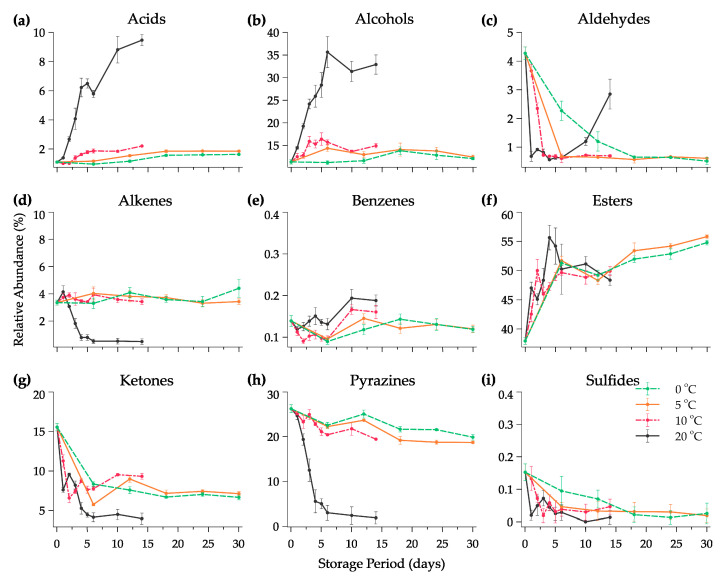
Changes in the relative abundance of volatile compounds (**a**–**i**) as functional groups in halal Kimchi under different temperature conditions over a storage period. Note: Data are expressed as mean ± standard deviations; *n* = 6.

**Figure 5 foods-11-02323-f005:**
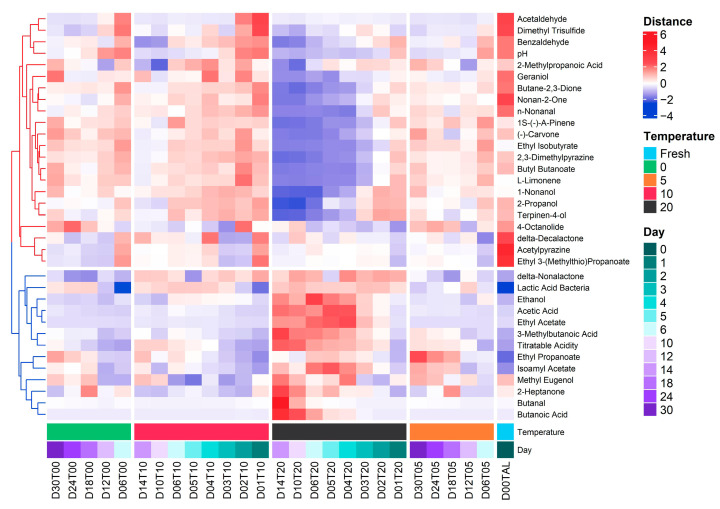
Heatmap representing the volatile compounds (VOCs) composition and quality indexes (pH, TA, and LAB) among different storage conditions (temperature (°C) and time (day)) of halal-certified Kimchi products.

**Figure 6 foods-11-02323-f006:**
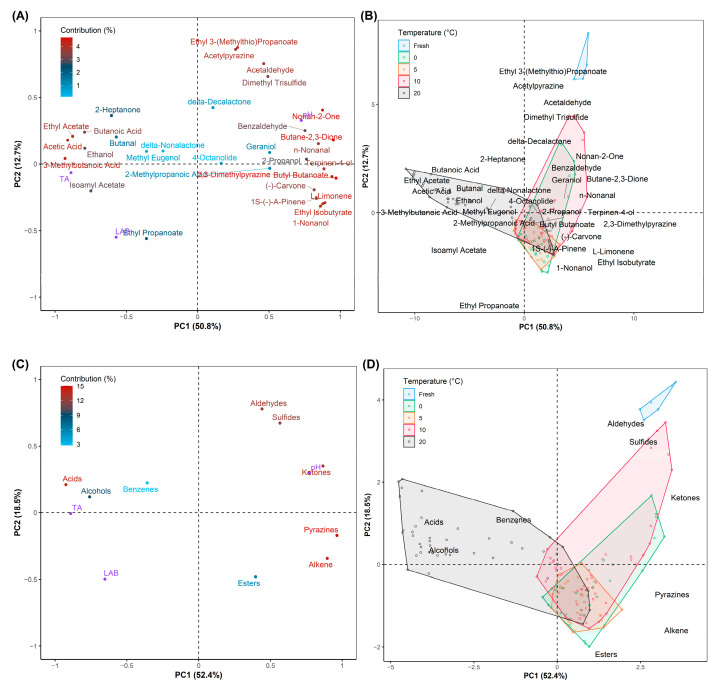
Relevant loadings of (**A**) volatile compounds and (**C**) functional groups of VOCs with additional information of quality indexes as predicted contributors (purple color) on principal component analysis (PCA); PCA-biplot of (**B**) volatile compounds and (**D**) functional groups of VOCs of halal-certified Kimchi over storage time.

**Table 1 foods-11-02323-t001:** Volatile compound composition and odor descriptor of fresh Kimchi.

Chemical Compounds	CAS	Formula	Relative Abundance (%) ^1^	Odor Descriptor ^2^
Acetaldehyde	75-07-0	C_2_H_4_O	3.44 ± 0.22	Pungent, aldehydic fruity
Ethanol	64-17-5	C_2_H_6_O	8.26 ± 0.39	Alcoholic, ethereal
Butanal	123-72-8	C_4_H_8_O	0.03 ± 0.02	Pungent, musty, malty
Butane-2,3-Dione	431-03-8	C_4_H_6_O_2_	1.41 ± 0.07	Butter, sweet, creamy
Ethyl Acetate	141-78-6	C_4_H_8_O_2_	0.00 ± 0.00	Ethereal, fruity, sweet
Acetic Acid	64-19-7	C_2_H_4_O_2_	0.10 ± 0.03	Sour, vinegar
Ethyl Propanoate	105-37-3	C_5_H_10_O_2_	2.61 ± 0.07	Sweet, fruity, rum
Ethyl Isobutyrate	97-62-1	C_6_H_12_O_2_	27.72 ± 0.55	Sweet, ethereal, fruity, rummy
2-Methylpropanoic Acid	79-31-2	C_4_H_8_O_2_	0.24 ± 0.01	Acidic, sour, rancid
3-Methylbutanoic Acid	503-74-2	C_5_H_10_O_2_	0.75 ± 0.03	Sour, stinky
2-Heptanone	110-43-0	C_7_H_14_O	0.12 ± 0.02	Fruity, spicy, sweet
2,3-Dimethylpyrazine	5910-89-4	C_6_H_8_N_2_	26.22 ± 0.90	Nutty, caramellic, roasted
1S-(-)-Alpha-Pinene	7785-26-4	C_10_H_16_	1.92 ± 0.13	Sharp, resinous, fresh, pine
Benzaldehyde	100-52-7	C_7_H_6_O	0.31 ± 0.01	Strong, sweet, bitter
Butyl Butanoate	109-21-7	C_8_H_16_O_2_	2.12 ± 0.26	Fruity, green
Nonan-2-One	821-55-6	C_9_H_18_O	12.08 ± 0.41	Fresh, sweet, herbal
1-Nonanol	143-08-8	C_9_H_20_O	0.56 ± 0.05	Floral, oily
(-)-Carvone	6485-40-1	C_10_H_14_O	0.14 ± 0.01	Sweet, herbal, minty
Methyl Eugenol	93-15-2	C_11_H_14_O_2_	0.14 ± 0.01	Spicy, clove, cinnamon
2-Propanol	67-63-0	C_3_H_8_O	1.50 ± 0.06	Alcohol, musty
Isoamyl Acetate	123-92-2	C_7_H_14_O_2_	0.64 ± 0.03	Sweet, banana
Butanoic Acid	107-92-6	C_4_H_8_O_2_	0.00 ± 0.00	Acetic, butter, fruit
Dimethyl Trisulfide	3658-80-8	C_2_H_6_S_3_	0.15 ± 0.03	Sulfurous, onion, meaty
L-Limonene	5989-54-8	C_10_H_16_	1.43 ± 0.08	Terpene, pine, peppery
Acetylpyrazine	22047-25-2	C_6_H_6_N_2_O	0.90 ± 0.22	Nutty, corn chip, hazelnut
n-Nonanal	124-19-6	C_9_H_18_O	0.49 ± 0.01	Aldehydic, rose, orange peel, fatty
Ethyl 3-(Methylthio)Propanoate	13327-56-5	C_6_H_12_O_2_S	4.83 ± 0.43	Sulfury, pineapple, ripe tomato
Terpinen-4-ol	562-74-3	C_10_H_18_O	0.84 ± 0.05	Pepper, musty, sweet
Geraniol	106-24-1	C_10_H_18_O	0.16 ± 0.12	Sweet, rose, citrus
4-Octanolide	104-50-7	C_8_H_14_O_2_	0.09 ± 0.01	Creamy, dairy, fatty
delta-Nonalactone	3301-94-8	C_9_H_16_O_2_	0.58 ± 0.16	Creamy, sweet, milky, coumarin
delta-Decalactone	705-86-2	C_10_H_18_O_2_	0.22 ± 0.09	Oily, coconut, peach, creamy

^1^ Values represent the mean ± SD; *n* = 6. ^2^ The odor descriptors of chemical compounds were confirmed and identified from the AroChemBase library (Alpha MOS, France) and the online website of TGSC Information System.

**Table 2 foods-11-02323-t002:** Volatile compound composition and odor descriptor of fresh Kimchi.

RT-Col	Volatile Compounds	Intensity (pA) *				
		Fresh	0 °C	5 °C	10 °C	20 °C
14.69-1	Acetaldehyde	5936.05 ± 458.87 ^a^	230.38 ± 18.16 ^b^	267.98 ± 25.90 ^b^	319.75 ± 34.18 ^b^	414.00 ± 49.75 ^b^
15.84-1	Ethanol	14236.46 ± 505.40 ^c^	19123.28 ± 588.82 ^b^	20397.39 ± 696.04 ^b^	18108.80 ± 711.05 ^b^	43380.52 ± 4352.83 ^a^
19.55-1	Butanal	51.32 ± 36.63 ^b^	87.50 ± 42.21 ^b^	89.32 ± 4.09 ^b^	10.20 ± 22.81 ^b^	3252.23 ± 772.99 ^a^
20.67-1	Butane-2,3-Dione	2421.72 ± 66.69 ^a^	1440.44 ± 8.70 ^b^	1527.13 ± 59.72 ^b^	1196.84 ± 45.67 ^c^	315.37 ± 22.75 ^d^
21.66-1	Ethyl Acetate	0.00 ± 0.00 ^c^	982.39 ± 116.31 ^b,c^	873.22 ± 59.88 ^b,c^	3457.81 ± 657.24 ^b^	44498.85 ± 3420.19 ^a^
22.71-1	Acetic Acid	169.06 ± 51.71 ^c^	758.39 ± 65.31 ^b^	902.61 ± 43.20 ^b^	905.43 ± 44.45 ^b^	6322.28 ± 370.90 ^a^
30.36-1	Ethyl Propanoate	4496.92 ± 64.51 ^e^	24185.36 ± 1118.86 ^b^	31314.76 ± 927.59 ^a^	21677.37 ± 585.27 ^c^	16395.09 ± 276.79 ^d^
36.29-1	Ethyl Isobutyrate	47828.95 ± 1890.70 ^b^	77147.01 ± 2611.94 ^a^	73947.33 ± 1596.34 ^a^	43734.76 ± 1530.32 ^c^	1856.62 ± 1899.21 ^d^
38.72-1	2-Methylpropanoic Acid	417.38 ± 9.85 ^a^	436.64 ± 11.03 ^a^	437.11 ± 16.57 ^a^	320.81 ± 11.35 ^b^	291.57 ± 7.98 ^c^
48.59-1	3-Methylbutanoic Acid	1297.08 ± 25.56 ^d^	2023.12 ± 83.52 ^c^	2399.99 ± 92.96 ^b^	2044.82 ± 75.31 ^c^	4271.11 ± 302.68 ^a^
49.79-1	2-Heptanone	206.60 ± 42.03 ^b^	137.10 ± 22.32 ^c^	147.61 ± 23.96 ^c^	139.64 ± 13.59 ^c^	382.52 ± 27.09 ^a^
54.27-1	2,3-Dimethylpyrazine	45216.99 ± 1681.18 ^a^	39449.12 ± 1939.13 ^b^	37925.16 ± 874.62 ^b^	28931.37 ± 516.02 ^c^	2458.73 ± 1552.58 ^d^
55.88-1	1S-(-)-Alpha-Pinene	3303.28 ± 159.95 ^b^	4521.97 ± 1266.03 ^a^	3543.64 ± 598.88 ^a,b^	2957.95 ± 328.49 ^b^	312.38 ± 142.11 ^c^
57.72-1	Benzaldehyde	537.80 ± 15.26 ^a^	359.56 ± 14.09 ^b^	355.78 ± 9.81 ^b^	256.82 ± 35.76 ^c^	232.85 ± 25.19 ^c^
59.26-1	Butyl Butanoate	3660.01 ± 487.85 ^a^	3899.66 ± 232.79 ^a^	3442.96 ± 376.10 ^a^	2116.45 ± 76.66 ^b^	369.42 ± 79.96 ^c^
67.70-1	Nonan-2-One	20836.86 ± 819.51 ^a^	9703.37 ± 198.14 ^c^	10707.08 ± 288.13 ^b^	10221.10 ± 499.13 ^b,c^	2677.87 ± 512.02 ^d^
71.04-1	1-Nonanol	973.50 ± 62.37 ^b^	1243.10 ± 51.33 ^a^	1228.38 ± 60.08 ^a^	995.57 ± 33.86 ^b^	432.09 ± 83.59 ^c^
75.77-1	(-)-Carvone	233.77 ± 9.65 ^b^	256.92 ± 16.95 ^a^	253.94 ± 13.32 ^a,b^	186.92 ± 8.28 ^c^	131.12 ± 10.77 ^d^
82.62-1	Methyl Eugenol	238.98 ± 16.89	235.71 ± 15.86	243.04 ± 18.94	238.67 ± 20.32	257.31 ± 14.29
20.79-2	2-Propanol	2581.57 ± 64.34 ^a^	1991.84 ± 55.59 ^b^	2128.22 ± 93.08 ^b^	1850.65 ± 236.01 ^b^	952.22 ± 443.22 ^c^
51.60-2	Isoamyl Acetate	1104.84 ± 20.97 ^c^	1684.74 ± 37.74 ^b^	2030.85 ± 112.12 ^a^	1714.66 ± 88.82 ^b^	1936.43 ± 46.64 ^a^
53.24-2	Butanoic Acid	0.00 ± 0.00 ^b^	0.00 ± 0.00 ^b^	0.00 ± 0.00 ^b^	0.00 ± 0.00 ^b^	2119.24 ± 266.194 ^a^
61.31-2	Dimethyl Trisulfide	263.33 ± 47.06 ^a^	50.65 ± 57.65 ^b^	38.08 ± 38.56 ^b^	68.87 ± 31.51 ^b^	18.22 ± 25.83 ^b^
63.07-2	L-Limonene	2464.58 ± 159.28 ^c^	4202.90 ± 215.28 ^a^	3403.25 ± 436.42 ^b^	2139.14 ± 135.23 ^c^	337.38 ± 76.67 ^d^
66.47-2	Acetylpyrazine	1559.91 ± 389.26 ^a^	75.91 ± 7.96 ^b^	159.80 ± 13.99 ^b^	185.31 ± 30.97 ^b^	121.44 ± 16.37 ^b^
71.22-2	n-Nonanal	840.39 ± 24.93 ^a^	331.88 ± 234.73 ^b^	517.22 ± 12.99 ^b^	444.43 ± 17.56 ^b^	27.87 ± 39.42 ^c^
72.72-2	Ethyl 3-(Methylthio)Propanoate	8339.19 ± 746.77 ^a^	940.34 ± 56.81 ^c^	1673.09 ± 63.54 ^b^	1690.49 ± 267.59 ^b^	1288.39 ± 93.11 ^b,c^
74.99-2	Terpinen-4-Ol	1455.39 ± 92.18 ^a^	1348.82 ± 72.48 ^a,b^	1290.71 ± 69.85 ^b^	1065.65 ± 79.68 ^c^	448.54 ± 83.72 ^d^
79.38-2	Geraniol	268.86 ± 179.21	264.48 ± 252.68	124.23 ± 24.92	188.85 ± 217.06	0.00 ± 0.00
85.52-2	4-Octanolide	157.36 ± 15.04 ^a^	153.27 ± 11.64 ^a,b^	148.16 ± 9.41 ^a,b,c^	130.51 ± 6.65 ^c^	136.58 ± 10.28 ^b,c^
90.99-2	delta-Nonalactone	1005.11 ± 240.42 ^b^	1198.01 ± 196.52 ^a,b^	1289.37 ± 230.15 ^a,b^	1537.97 ± 139.35 ^a^	1475.41 ± 272.36 ^a^
94.00-2	delta-Decalactone	368.51 ± 122.67	216.28 ± 190.43	202.11 ± 72.95	271.56 ± 163.85	175.87 ± 82.49

Note: All data are expressed as mean ± SD with different lowercase letters (a–d) in the same row are significantly different (*p* < 0.05), *n* = 6. Abbreviations: RT, Retention Time; Col, Column Number of e-nose.

## Data Availability

All data presented within the article are available at request from corresponding author.

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
