# Peer review of "Real-Time Monitoring the Effects of Storage Conditions on Volatile Compounds and Quality Indexes of Halal-Certified Kimchi during Distribution Using Electronic Nose"

_foods, 2022, doi:10.3390/foods11152323_

Round 1

Reviewer 1 Report

The electronic nose and the conventional method with the integration of multivariate analysis are used to study the effect of temperature on volatile compounds, quality indexes, and shelf life of Halal-certified Kimchi, and to identify alcohol and find a correlation between the variables. This article has an interesting subject, and the purpose of the work has been appropriately presented. The topics presented are important from a scientific as well as practical perspective in the area of measurement in food production and processing engineering. The manuscript is well written. However, there are some points that need to be considered as follows:

1- The abstract and conclusion need to be revised to include results.

2- The authors should describe the steps of signal processing. How to deal with the original signals and how to select the signals ?? The details about signal treatment should be given.

3- The difference between the e-nose response and the conventional method is not clear and it needs to be clarified in order to see the results of the e-nose test.

4- The response of the e-nose sensors should be provided.

5- Line 308:  "PC2 185% are presented in" should be replaced with 18.5%.

6- There is only a brief discussion of the results obtained in the paper and it should be thoroughly revised.

7- Figure 4 needs to be discussed further since it is unclear, and I think the distance term is inappropriate and should be replaced with correlation.

Reviewer 2 Report

This paper focuses on Real-Time Monitoring the Effects of Storage Conditions on Volatile Compounds and Quality Indexes of Halal- Certified Kimchi during Distribution using Electronic Nose. The topic is meaningful and interesting. Odor Descriptor has also been clearly presented in Tables, which is important. I have only a few comments to improve this work.

 In Introduction section, several techniques have been mentioned. In my opinion, several fast spectroscopic methods should also be included to give a global introduction. For example, near-infrared spectroscopy, hyperspectral imaging (https://doi.org/10.1016/j.infrared.2022.104098, https://doi.org/10.1016/j.infrared.2022.104169), etc, these techniques are quite hot in recent years.

 Line 80,“to volatile compounds should be modified.

 Section 2.2, Have you conducted measurements according to reference methods to determine pH, TA, LAB, Volatile Compounds, etc? Please add some references.

 Section 3.3.2, PCA should be described in M&M section.

Reviewer 3 Report

The manuscript deals with the evaluation of a traditional Korean fermented food called Kimchi according to its volatile composition using an electronic nose and the variation of the composition of these substances and other parameters during the storage process at different temperatures. The methodology employed is in accordance with the objectives of the work, which were achieved and duly discussed in view of the results obtained. The work needs some minor modifications, which I describe below:

Line 92: "at 0ºC and 4ºC", I believe is "at 0ºC and 5ºC";

Line 178: Shelf should be written in lower case

Fig.2: the gray line should be indicated in the caption as the evolution of the internal temperature during the experiment

Line 250: Extreme should be written in lower case

Line 308: "PC2 185%" I believe is "PC2 18.5%"

Round 2

Reviewer 1 Report

The authors have revised the paper and addressed the comments properly. I recommend that the paper be accepted.